# Relationship between Physicochemical and Cooking Quality Parameters with Estimated Glycaemic Index of Rice Varieties

**DOI:** 10.3390/foods13010135

**Published:** 2023-12-30

**Authors:** Cristiana L. Pereira, Inês Sousa, Vanda M. Lourenço, Pedro Sampaio, Raquel Gárzon, Cristina M. Rosell, Carla Brites

**Affiliations:** 1National Institute for Agricultural and Veterinary Research (INIAV), I.P., Av. da República, 2780-157 Oeiras, Portugal; 2Department of Earth Sciences, NOVA University of Lisbon, 2829-516 Caparica, Portugal; 3Linking Landscape, Environment, Agriculture and Food (LEAF) Research Center, Tapada da Ajuda, 1349-017 Lisboa, Portugal; 4Center for Mathematics and Applications (NOVA Math) and Department of Mathematics, NOVA SST, 2829-516 Caparica, Portugal; 5Computação e Cognição Centrada nas Pessoas, BioRG—Biomedical Research Group, Lusófona University, Campo Grande, 376, 1749-019 Lisboa, Portugal; 6Institute of Agrochemistry and Food Technology (IATA-CSIC), 46980 Paterna, Spain; 7Food and Human Nutritional Department, University of Manitoba, Winnipeg, MB R2H 2A6, Canada; 8GREEN-IT Bioresources for Sustainability, ITQB NOVA, Av. da República, 2780-157 Oeiras, Portugal

**Keywords:** rice commercial types, physicochemical parameters, cooking parameters, glycaemic index

## Abstract

Rice is a significant staple food in the basic diet of the global population that is considered to have a high glycaemic index. The study of the physical and chemical parameters in rice that are related to the starch digestion process, which allows us to quickly predict the glycaemic index of varieties, is a major challenge, particularly in the classification and selection process. In this context, and with the goal of establishing a relationship between physicochemical properties and starch digestibility rates, thus shedding light on the connections between quality indicators and their glycaemic impact, we evaluated various commercial rice types based on their basic chemical composition, physicochemical properties, cooking parameters, and the correlations with digestibility rates. The resistant starch, the gelatinization temperature and the retrogradation (setback) emerge as potent predictors of rice starch digestibility and estimated glycaemic index, exhibiting robust correlations of r = −0.90, r = −0.90, and r = −0.70 (*p* ≤ 0.05), respectively. Among the rice types, Long B and Basmati stand out with the lowest estimated glycaemic index values (68.44 and 68.10), elevated levels of resistant starch, gelatinization temperature, and setback values. Furthermore, the Long B showcases the highest amylose, while the Basmati with intermediate, revealing intriguingly strong grain integrity during cooking, setting it apart from other rice varieties.

## 1. Introduction

Rice holds a significant position among cereal crops, serving as a crucial energy source and being widely integrated into the diets of populations. Ensuring the quality of rice grains is of paramount importance due to its impact on consumer preferences, acceptance, and its influence on economic value [1]. The evaluation of grain quality encompasses a spectrum of factors, including its physical attributes (such as form, size, and colour), its fundamental chemical composition (including protein, lipid, and fibre content), and its physicochemical characteristics (such as amylose content, gelatinization temperature and viscosity profiles). The cooking and eating quality of rice is directly affected by amylose content with impact on the water absorption during cooking [2], in the grain elongation ratio, in the final hardness of the cooked grain [3], in the gelatinization temperature and in the rate of retrogradation [4].

Rice, being a starchy food, is considered a high-glycaemic index (GI) food [5]. The prevalence of diabetes mellitus (DM) and its projected increase to 10.2% of the world population by 2030 [6] pose a significant challenge to the entire rice value chain. As a result, selecting rice varieties with a low GI is now a priority in rice production, along with promoting the consumption of brown rice. 

The determination of the GI of food is standardized through an in vivo procedure [7], but in vitro methods for evaluating the estimated glycaemic index (eGI) based on the starch digestibility rate are more convenient and rapid [8,9]. Considering its digestibility rate, starch can be classified into three types: rapidly digestible starch (RDS), which is digested within 20 min; slowly digestible starch (SDS), which takes about 120 min to digest; and resistant starch (RS), which remains undigested [8]. RS is resistant to hydrolysis by the enzymes α-amylase and α-amyloglucosidase in the small intestine. It instead reaches the large intestine, where it undergoes fermentation by colonic microflora [8,10]. Rice grains containing over 2% of RS exhibit lower GI values (≤60) [10]. Other in vitro methods for indirectly assessing eGI include the quantification of analytes in the glucometer [11,12] or measuring the impact of amylolytic enzymes on apparent viscosity [13,14], by using a rapid viscoanalyzer (RVA). 

Although rice is considered a high GI food, the genetic background of the different varieties results in notable diversity [15]. Indeed, enzymes involved in the amylose synthesis during grain development influence the GI [15]. Additionally, the inherent characteristics of each variety can impact glycaemic response [16]. 

The ratio of amylose to amylopectin in starch, along with other molecular structural characteristics, determines its ultimate digestibility [17] with starches with higher amylose content exhibiting greater resistance to digestion [18]. In addition, the starch biosynthetic pathway can influence the distribution of branch (chain) lengths in both amylose and amylopectin, consequently affecting the GI [17,19]. The genome studies focusing on starch biosynthesis have been employed in rice to alter soluble starch synthase isoforms, resulting in an enhancement of amylose content up to 63% and an increase in resistant starch [20]. 

Apart from starch composition, additional macronutrients found in rice, such as protein [21], fibre and lipid contents, can impact starch digestibility [22]. The formation of complexes with starch [23] could potentially reduce its vulnerability to amylolytic enzymes and limit starch gelatinization. The information available highlights the variability among rice varieties concerning their physicochemical properties and cooking performance. Moreover, there is an established connection between certain rice properties with GI. Despite this, no prior study has centred on employing physicochemical and digestibility analyses to effectively characterize and differentiate rice varieties.

The principal objective of this study is to establish a relationship between the physicochemical attributes and cooking properties of rice and its starch digestibility rates. To achieve this, we evaluated 22 rice varieties cultivated in the Mediterranean Region, categorized based on their culinary or commercial type. Our assessment encompassed (i) the fundamental chemical composition (protein, fibre and lipids); (ii) physicochemical parameters (amylose content, gelatinization temperature, and viscosity profiles); (iii) cooking characteristics (water absorption, volumetric expansion ratio and solids leached); and (iv) starch digestion parameters obtained via two distinct in vitro methods: enzymatic procedure (eGI, RDS, SDS, RS, and TS) and RVA digestograms (kinetic constant, kRVA). This study encompassed an evaluation of the variability across rice varieties and commercial types, while also calculating correlations between parameters to identify predictors of the glycaemic index. 

## 2. Materials and Methods

### 2.1. Materials

For this study, 22 brown rice varieties were selected; 20 were cultivated in the Mediterranean Region (including *Indica* and *Japonica* subspecies) from Europe and Egypt (Giza 177, Giza 181) and 2 were imported outside Europe as Basmati varieties. The 22 varieties were classified according to their commercial type: Long A, Medium grain, Short grain, Long B, European aromatic and Basmati types (Table 1). The commercial classification of grains relied on their kernel biometry following the Codex Standard 198-1995 [24] for rice. Alongside grain biometry, rice types were also categorized based on flavour characteristics, including varieties such as Basmati and European aromatic types. 

### 2.2. Rice Samples Preparation

The brown grains were polished (Suzuki MT-98, Santa Cruz do Rio Pardo, São Paulo, Brazil) to produce milled grains. The rice brown flours (moisture content ~12%) were obtained by using a Cyclone Sample Mill (Falling number 3100, Perten, Stockholm, Sweden), with a 0.8 mm screen and were used in all analyses, except for the amylose quantification in what was adopted a standardized methodology applied to milled rice. The cooking parameters were determined also on milled grains and a sample from each rice variety (12 g) was mixed with 120 g of distilled water and was cooked for 20 min (a fixed duration for all varieties). 

### 2.3. Chemical Basic Composition

Protein (PRTD), fibre (FIBER) and fat (FAT) contents: The fat, fibre and protein contents were determined using the NIR (Near-Infrared Spectroscopy) method in brown rice flours with MPA equipment (Bruker Optics, Ettlingen, Germany) using the cereals B-FING package calibration, provided by Bruker Company (MA, USA). 

### 2.4. Physicochemical Parameters

Amylose content (AMYL): The amylose content was determined by a spectrophotometric method for milled rice samples, according to EN ISO 6647-2:2020 [25]. The absorbance was measured at 720 nm and then was used with the calibration graph to determine the amylose content.

Pasting properties and gelatinization temperature: The determination of the pasting properties of rice with the RVA (Rapid Visco Analyser, (Newport Scientific Pty Ltd., Warriewood NSW 2012, Australia) equipment was conducted according to the AACC 61-02.01 method. Each rice brown flour was mixed with distilled water and measured on a dry weight (dw) basis. The slurries were subjected to a cycle of heating and cooling under continuous stirring. Values such as peak viscosity (VPEAK) and the setback (SB-difference between final viscosity and VPEAK) were obtained from the viscosity curve. The gelatinization temperature (Tg) of the rice samples was obtained according to the AACC 61-04.01 method. 

### 2.5. Cooking Parameters 

Cooked rice was analysed according to the method described by Ferreira et al. (2017) [26] with some modifications concerning the amount of sample and using an electric pan, to obtain the water uptake ratio (WUp) and volumetric expansion ratio (VER). The cooking water was analysed for solids leached (*SL*), calculated according to Altheide et al. (2012) [27]. 

Water uptake (WUp): immediately after cooking, the rice sample was drained during 5 min and weighed. The WUp was calculated based on weight (*g*) gained during cooking (cooked rice weight) using Equation (1): (1)$$\mathrm{WUp}=\frac{cooked\;rice\;\left(g\right)-uncooked\;rice\;(g)}{uncooked\;rice\;(g)}\times \;100$$

The Volumetric expansion ratio (VER) was determined through the ratio of the volume of cooked rice to the volume of uncooked rice (raw rice), as in the following equation (Equation (2)):*VER* = *cooked rice* (*mL*)/*uncooked rice* (*mL*)(2)

During cooking, weight variations are not a measure of water absorption alone; rice may also leach out starch and other soluble solids that can be quantified by solids leached (*SL*).

Solids leached (SL) was evaluated by drying an aliquot (50 mL) of cooking water, at a temperature of 102 °C, which was evaporated during 24 h in a glass container. *SL* was measured by the difference of weight of the glass container with dry aliquot (W1) and the weight of the empty container (W2), using the following equation (Equation (3)):(3)$$SL=\frac{W1\;(g)-W2\;(g)}{uncooked\;rice\;(g)}\;\times \;100$$

### 2.6. Starch Digestion Parameters by In Vitro Methods

#### 2.6.1. In Vitro Enzymatic Method (eGI, RDS, SDS, RS and TS) 

The rapid and slowly digestible starch (RDS and SDS), the resistant starch (RS) and the total starch (TS) fractions were evaluated using a starch assay kit (K-DSTRS) from Megazyme (Wicklow, Ireland) according to Englyst et al. (1992) [8]. The kinetics of starch digestion and estimated GI were calculated according to the non-linear model earlier described by Gõni et al. (1997) [9], expressed by Equations (4) and (5).
*C* = *C*_∞_(1 − *e*^−*kt*^)(4)
(5)$$\mathrm{AUC}=C_{\infty }(t_{f}-t_{0})+\frac{C\infty }{k}(e^{-kt_{f}}-e^{-kt_{0}})$$
where *C* corresponds to the percentage of starch hydrolysed at *t* time; *C_∞_* is the equilibrium concentration of starch hydrolysed after 240 min, *k* is the kinetic constant and *t* is the time (min). The hydrolysis index (*HI*) was obtained by dividing the estimated areas under the starch hydrolysis curve, *AUC* sample by *AUC* reference sample (glucose). The predicted glycaemic index (*GI*) was calculated using a model (Equation (6)): *GI =* 39.6207 *+* (0.5498 *HI*)(6)

The starch digestion parameters were determined in raw brown rice but in a preliminary study, in order to understand if the behaviour of the glycaemic index of varieties remained the same after cooking, 12 varieties were selected for assessing *GI* after cooking. The brown grains were cooked for 60 min at a ratio of 16 g of rice to 48 g of distilled water, and then lyophilized and milled for analysis.

#### 2.6.2. RVA Digestograms (Kinetic Constant, k_RVA_)

Starch hydrolysis was determined by means of a rapid methodology developed by Santamaria et al. (2022) [13] that linked the measurement of rice flour performance with the prediction of starch digestibility. Briefly, 3 g (14% moisture basis) of starch were dispersed in 25 mL distilled water and placed into the RVA canister using the following settings: 50 °C for 1 min, heating from 50 to 95 °C at 10 °C min^−1^, holding at 95 °C for 2.5 min, and cooling down to 37 °C at 10 °C min^−1^. For the digestogram stage, the temperature was hold at 37 °C for 36 s for adding the α-amylase solution (90 U 100 µL^−1^ that represented 30 U g^−1^ of starch) and then the recording of apparent viscosity continued for 5 min. Rotational speed in the first 10 s was 960 rpm and then it was kept at 160 rpm along the test. Pasting parameters extracted included the initial (after adding the enzyme), and final (at the end of the assay) viscosity during the enzymatic hydrolysis. A first-order kinetic model was applied to model the digestograms, (Equation (7)):*μ* = *μ*_∞_ + (*μ*_0_ − *μ*_∞_)*e^−kt^*(7)
where *µ* is the apparent viscosity (mPa s), *µ*_0_ is the initial viscosity, *µ*_∞_ is the final viscosity, *k* (min^−1^) is the kinetic constant, and *t* (min) is the hydrolysis time.

### 2.7. Statistical Analysis Data

The variability of data among different commercial types of rice was depicted using boxplots, and the analysis included a one-way ANOVA with mean comparisons conducted using the Tukey high significant differences test (HSD) by R software (version 4.3.1.) [28]. Here, we assume that all the assumptions underlying the hypothesis tests are met, given the limited number of observations in each comparison group, making it impractical to validate them thoroughly. This also implies that all results need to be interpreted in light of the small sample size. In particular, significant associations might have gone undetected. Significant correlations between all the parameters were determined with Pearson correlations analysis and a test for association between paired samples. To further explore the relationships among variables, a principal component analysis (PCA) was conducted. Statistical analyses were carried out with a significance level of 5%. The classification toolbox for MATLAB^®^ (version 2.0) was utilized to perform principal component analysis (PCA). MATLAB^®^ 2018 [29] was employed for all pre-processing and analysis methods. The R software was used to perform all the analyses as well as model the first-order kinetic equations for glycaemic index estimation.

## 3. Results and Discussion

The 22 varieties were sorted into six distinct rice commercial types (Long A, Long B, Short grain, Medium grain, European Aromatic, and Basmati), and the raw data results are available at https://doi.org/10.34636/DMPortal/5G5DA5 (accessed on 29 December 2023). The varieties categorized into the commercial types Long A, Short grain, and Medium grain belong to the *Japonica* subspecies, while Long B belongs to the *Indica* subspecies. This classification significantly influences the results of the analysed parameters. 

For this investigation, we examined the digestibility and nutritional parameters in samples of brown rice. The decision to use brown rice rather than milled rice was justified by its lower carbohydrate content and its richer nutritional profile in terms of lipids, proteins, and dietary fibre—compounds that exert a more pronounced impact on glycaemic index [22].

Apart from its basic nutritional composition, the bran found in brown rice is abundant in bioactive compounds such as γ-oryzanol, ferulic acid, phytic acid, γ-aminobutyric acid, tocopherols, and tocotrienols (vitamin E). The presence and levels of these compounds vary depending on rice variety and types, underscoring potential therapeutic effects on metabolisms associated with diabetes. Certain effects directly impact the glycaemic index, such as the inhibitory action of phytic acid on starch-digesting enzymes, as emphasized by Pereira et al. (2021) [22] and Tuaño et al. (2021) [30].

The results of starch digestion parameters, chemical composition, physicochemical and cooking parameters are represented by their respective boxplots (Figure 1). 

In order to understand the variability between the different rice commercial types, we compared the means of the starch digestion, chemical composition, physicochemical and cooking parameters (Table 2). The European Aromatic type was not included in the group means comparison due to the availability of only one observation.

### 3.1. Chemical Composition and Viscosity 

In terms of chemical basic composition, significant differences between two rice types were observed in protein (PRTD) and fibre (FIBER) contents, as evidenced by values ranging between 6.99% (Medium grain type) and 9.09% (Basmati type) for PRTD, and between 1.12% (Medium grain type) and 1.94% (Basmati type) for FIBER, as displayed in Table 2. Notably, in this study the FIBER and PRTD content also differed significantly between the Basmati (1.94% and 9.09%) and Medium grain types (1.12% and 6.99%). However, no significant differences were observed in the fat (FAT) content among the different rice types. 

Regarding the physicochemical composition, specifically the amylose (AMYL) content, significant differences were detected between the amylose contents of Long A (19.66%) and Long B (27.46%) types of rice. The difference in AMYL content was clearly evident, with *Indica* varieties (Long B) exhibiting a higher concentration compared to the intermediate levels observed in *Japonica* varieties (Long A).

Concerning the viscosity parameters, significant differences in Tg and SB parameters were noted between the Basmati and Long B rice varieties and the other three rice groups (i.e., Long A, Short grain and Medium grain). In this context, Tg and SB values ranged from 63.81 °C to 73.37 °C and 262 cP to 2318 cP, respectively, with Long B and Basmati types yielding the highest Tg and SB values. No significant differences were detected in the peak of viscosity (VPEAK) across the various rice types.

### 3.2. Cooking Parameters

Cooking under identical conditions enables us to comprehend the characteristics of distinct grain types, including their water absorption capacity, expansion volume, and resulting grain softening. The greatest water uptake ratio (WUp) was found for Basmati rice (246.9%), followed by Medium grain rice (224.2%), with values that were not deemed significantly different from one another. However, significant differences were observed between the Basmati (246.9%) and Long A (187.3%) rice varieties for this cooking parameter. While cooking rice, starch and other soluble solids may leach out. Hence, weight variation is not solely indicative of water absorption. The Solids leached (SL) parameter can quantify the loss of these substances accurately in particular, significant differences in SL content were observed between the Basmati (4.95%) and Medium grain (7.50%) types of rice.

### 3.3. Starch Digestion Parameters

Nowadays, the assessment of glycaemic response is becoming increasingly significant for preventing and controlling nutrition-related diseases, especially diabetes [31]. In vitro estimation of glycaemic index (eGI), calculated using the hydrolysis index (HI) [32], serves as a valuable nutritional useful tool, from the nutritional point of view, for accessing starch digestibility in foods based on glucose released.

Starch hydrolysis index is obtained through RDS, SDS, total starch (TS) and resistant starch parameters (RS) presented in Table 2. Table 2 indicates that there were no significant differences detected in digestible starch RDS and SDS contents, as well as in total starch (TS) content, among any of the rice types. However, regarding resistant starch (RS), significant differences were observed. Namely, between Long A (0.43%) and both Long B (4.28%) and Basmati (2.57%), as well as between Long B (4.28%) and both Short grain (0.32%) and Medium grain (0.11%).

The estimated glycaemic index (eGI) values ranged from 68.10 (Basmati type) to 75.88 (Medium grain type). The eGI results obtained in this study align with the reported range for rice, falling between medium and high GI [5]. The eGI values of Basmati and Long B varieties were notably lower, 68.10 and 68.44, respectively, compared to the Long A, Short and Medium grains varieties (Table 2). 

No significant differences were found among the Long A (74.33), Short grain (74.79) and Medium grain (75.88) rice varieties nor between the Long B (68.44) and Basmati (68.10) rice types. However, significant differences were detected in pairwise comparisons between any rice types within these two groups. Notably, the eGI for the first group of rice types was higher than that of the second group.

A rapid RVA model for analysing starch hydrolysis was designed and tested, through the digestograms of each rice variety. In relation to the hydrolysis rate (kRVA), it varied between 2.07 (Long B type) and 2.76 (Basmati type). Particularly, significant differences were identified between Basmati (2.76) and both Long A (2.26) with Long B (2.07) rice varieties.

### 3.4. Correlations between In Vitro Starch Digestion Parameters, Physicochemical, Viscosity and Cooking Parameters

The rate at which carbohydrates are digested in starchy products helps evaluate how glucose is released during consumption, which in turn contributes to in vitro evaluations of glycaemic responses. The glucose released after food consumption is influenced by several factors categorized into food and human elements. This study primarily focused on the impact of rice-related parameters on the eGI of rice varieties. 

The significant correlations between all pairs of parameters emphasize several important relationships (Figure 2).

Among the chemical composition FIBER and PRTD exhibited a positive correlation (r = 0.70) and were also the parameters demonstrating the strongest correlation with cooking parameters. Notably, FIBER correlated positively with the cooking parameter WUp (r = 0.60), while PRTD correlated negatively with the cooking parameter SL (r = −0.60). In particular, this means that varieties with higher FIBER content show higher WUp whereas varieties with higher protein content exhibit lower SL to the cooking water (Figure 2). TS constitutes the primary compound within the rice grain displaying, as expected, a significant negative correlation (r = −0.70) with PRTD. Basmati varieties stand out with the highest FIBER and PRTD contents (1.94% and 9.09%, respectively), while medium-grain varieties exhibit the lowest values (1.11% and 6.99%).

Authors who have evaluated the relationship between the basic chemical composition of certain starchy foods and their glycaemic index (GI) found that the correlation of values becomes more evident when assessing products of diverse types and amounts of starch compared to their proximate parameters, such as protein and fat contents. Furthermore, some studies reveal the formation of complexes with amylose (AMYL), which render the starch less susceptible to enzymatic hydrolysis [21,23]. Another contributing factor to lower GI is the denaturation and hydrolysis of proteins, which can impede starch hydration and enzymatic cleavage, thereby further reducing starch swelling [21].

The results obtained from the starch viscosity and gelatinization analysis using RVA are correlated with the VPEAK, SB and also the Tg of the brown rice flours. During gelatinization, the starch granules swell and hydrogen bonds are disrupted, leading to the breakdown of the granule structure, which reorganizes again during retrogradation [31]. The increase in viscosity with temperature may be attributed to the removal of water from the exuded AMYL by the granules as they swell. Granule swelling is accompanied by the leaching of granular constituents, predominantly AMYL, into the external matrix resulting in a dispersion of swollen granules in a continuous matrix. 

The SB shows the starch’s propensity for retrogradation tendency of the starch to retrograde, while in contrast to the viscosity peak, which shows when the starch reaches the maximum viscosity during heating (showing the stability of viscosity upon heating) illustrates the point at which the starch attains its highest viscosity during heating, denoting viscosity stability during this process [33]. The present study determined that VPEAK ranged between 1656 and 2537 cP, with Medium grain varieties displaying higher values, indicating a requirement for lower gelatinization temperatures. Nevertheless, the SB for this rice commercial type was lower, indicative of reduced final viscosity and hence greater susceptibility to digestion. Conversely, both Basmati and Long B varieties exhibited elevated Tg and SB values as well as lower digestibility compared to the other rice types. These findings demonstrated a strong correlation between Tg and SB with the eGI (r = −0.90 and r = −0.70, respectively), highlighting these variables as reliable predictors of GI.

The obtained correlation coefficients reveal interesting relationships between eGI and various parameters, including starch digestion parameters, as well as nutritional, cooking, and viscosity parameters measured in this study (Figure 2). Various factors could account for these differences. In terms of starch digestion parameters, significant differences were found for RS (*p* ≤ 0.05). Rice varieties displaying higher RS values tend to exhibit lower eGI (r = −0.90) and decreased RDS values (r = −0.70). These findings are consistent with assertions from several authors, affirming the direct impact of RS content on starch digestibility [10]. Furthermore, it was observed that samples with significant/high starch digestion parameter RDS content also exhibited higher eGI values (r = 0.90). RDS represents the starch fraction that is completely digested in the small intestine, contributing to an increase in blood glucose levels. On the contrary, RS comprises starch that is not absorbed in the small intestine but is instead fermented in the large intestine, offering benefits to colonic health and contributing to low starch digestibility [34]. Three parameters showed a negative correlation with eGI: RS from the starch digestion parameter group (r = −0.90), and SB and Tg from the viscosity parameter group (r = −0.70 and r = −0.90, respectively). 

Differences in the RVA hydrolysis rate (kRVA) among brown rice types were observed, with the Long B type displaying the lowest hydrolysis rate and the Basmati type showing the highest. Notably, this kinetic constant did not present a direct correlation with the eGI determined by enzymatic methods. However, it did show a significant and interesting correlation with the AMYL content (r = −0.50; Figure 2). In fact, the hydrolysis rate of rice was for Basmati rice (kRVA = 2.76), despite not having the highest AMYL content (19.93%). In contrast, the Long B variety exhibited the lowest hydrolysis rate (kRVA = 2.07) and a higher AMYL content (27.46%), making this model a strong predictor of AMYL content. 

The AMYL content showed a positive correlation with the viscosity parameter SB (r = 0.60). Additionally, there was a negative correlation between cooking parameter VER and viscosity parameter VPEAK (r = −0.60), indicating that varieties with a high VER tend to exhibit a low VPEAK.

While a direct correlation between AMYL content and eGI was not firmly established, given the intermediate AMYL values across most rice types, numerous studies have emphasized the interplay among starch digestibility, glycaemic response, and AMYL content [15,18]. 

The synthesis of starch, particularly regulated by specific genes, plays a pivotal role in determining various properties, including structure, functionality, and digestibility [17,35,36]. Genetic studies on rice germplasm collections have identified genes such as Wx, GBSSI, and SSIIa, which control high levels of resistant starch (RS) by being associated with elevated amylose levels [36,37]. Notably, rice with high GBSSI levels exhibits increased amylose content and higher levels of resistant starch [35,38,39]. Analyses of starch biosynthesis by Jabeen et al. (2021) revealed a higher accumulation of GBSSI and SSIIa proteins in low-glycaemic-index (GI) lines, contrasting with high GI lines.

The type of starch and its behaviour significantly impact the cooking process and, consequently, its digestibility [17,40,41]. While it is established that starches with higher amylose content are more resistant to digestion [18], researchers caution against relying solely on high amylose content as a predictor of glycaemic index (eGI). Further research has shown that long amylopectin chains in intermediate-AMYL rice starch resist enzymatic hydrolysis, similar to rapid retrogradation observed in amylose double helices [17,19,30,40,42,43]. The Basmati type serves as a notable example in this study showcasing intermediate AMYL content (19.93 ± 2.77) yet displaying a lower eGI, likely due to amylopectin chains contributing to higher RS (2.57 ± 0.06) and a distinctive rice texture compared to other intermediate-AMYL types (Long A). This characteristic promotes a separate, fluffy, and non-sticky consistency when cooked, exhibiting a low eGI.

The influence of compositional factors beyond amylose and amylopectin synthesis on eGI is underscored by additional research [30]. Varieties with high levels of amylose in the endosperm tend to be rich in RS type 3, demonstrating starch’s conversion into resistant starch via retrogradation after cooking, which is less susceptible to α-amylase digestion. Varied starch digestibility in rice samples is attributed to differences in amylose concentration and the content of long-chain amylopectin in intermediate and high amylose samples.

Pan et al. (2022) [43] found varying digestibility among six rice varieties with high AMYL, indicating a strong positive correlation between resistant starch (RS) with an intermediate chain amylopectin.

Interestingly, the correlation analysis, excluding the Basmati type in this study, reveals negative correlations between AMYL content and eGI, as well as between kRVA and SB—two variables intricately linked to eGI. This underscores the complexity of factors influencing glycaemic response, suggesting that a comprehensive understanding of rice types goes beyond individual components like AMYL content.

### 3.5. Principal Component Analysis (PCA)

The principal component analysis (PCA) was performed on the 16 variables including glycaemic, chemical composition, viscosity and cooking parameters. PCA was characterized by PC1 35.86% and PC2—35.83%, which could explain 71.69% of total variance (Figure 3). 

The PC1 represented the eGI, RDS, RS, and SB parameters. The second principal component (PC2) was mainly attributed to AMYL contents, and SL, PRTD, kRVA parameters. The distribution obtained from the principal components showed that rice varieties with lower eGI have lower digestion rates, and lower RDS contents and higher RS contents and Tg, as well as a greater tendency to retrogradation (SB). As expected, the relationships obtained from the PCA are in agreement with the previous correlations results. The distribution of the 22 varieties between the PC1,2 shows a clear separation between three groups of rice commercial types: 1—Long A, Medium, Short grain; 2—Long B; 3—Basmati. 

### 3.6. In Vitro Starch Digestibility Curves and eGI of Brown Rice Varieties 

The 22 varieties of raw brown rice, distributed among the 6 commercial types of rice, were individually analysed for starch digestibility using an in vitro method, obtaining a hydrolysis curve for each rice variety (Figure 4a), and the hydrolysis rate (HI) was calculated from estimated the eGI. 

However, as preliminary study and to understand the influence of rice cooking on eGI, 12 selected varieties were cooked, lyophilized and analysed in the same conditions in order to understand if varieties with lower HI, maintained lower values after cooking (Figure 4b).

The hydrolysis rate of starch exhibited an increase with prolonged cooking time [44], aligning with expectations due to the increased availability of starch following the breakdown of starch chains. In its raw form, the samples demonstrated a relatively low total starch hydrolysis rate at 20 min of digestion, ranging from 12.4% to 35.6% (Figure 4a). 

Conversely, the cooked rice samples exhibited considerably higher total hydrolysis rates, ranging from 46.3% to 59.6% within the same digestion period (Figure 4b). However, comparing the digestibility rates of raw or cooked rice the varieties order remained similar, the raw rice varieties with lower hydrolysis index (HI), such as CL28 (Long B rice), TipoIII and Super Basmati (Basmati rice’s), also maintain the lowest HI after cooking. Caravela, Ulisse and Arelate varieties (Long A types) showed higher HI before and after cooking.

The uniform cooking procedure provides insights into the diverse behaviours of various grain types, including water absorption capacity, expansion volume, and consequent grain softening [44].

It is noteworthy that the volumetric expansion ratio (VER) displays a positive correlation with the SB value (r = 0.70), while the SL shows a significant negative correlation with protein content (r = −0.60). These results align with previous literature [21] indicating that protein content inhibits starch hydration and subsequent leaching during the cooking process [23]. 

The glycaemic index was estimated for the 22 rice varieties in raw brown samples based on hydrolysis curves (Figure 5). 

As seen in Table 2, the Basmati and Long B rice types show lower eGI (68.10 ± 1.61 and 68.44 ± 2.59) compared with the other rice types (>70). The Long A type is the class with great number of varieties and denotes some variability among them (74.33 ± 1.85), with Albatros, Lusitano and Ronaldo standing out as varieties with the lowest eGI (Figure 5). 

## 4. Conclusions

In this study, we evaluated different commercial rice types in terms of their basic chemical composition, physicochemical and cooking properties, and also established relationships with digestibility rates. The Long B type displayed the highest amylose content, while both the Long B and Basmati types exhibited elevated levels of resistant starch and gelatinization temperature, along with lower eGI values. Additionally, we highlighted the distinct characteristics of the Basmati type, including its intermediate AMYL content and remarkable grain integrity during cooking. Long A, Short grain, and Medium grain types exhibit higher eGI, elevated viscosity, lower gelatinization temperatures, and resistant starch contents below 1%. Among them, medium-grain rice types displayed the least integrity during the cooking process. 

The resistant starch content, gelatinization temperature, and retrogradation (setback) parameters are recognized as indicators that can predict the digestibility rate of rice starch and its corresponding eGI. Given rice’s significance as a staple in the human diet, assessing its glycaemic index for quality classification and selection can serve as a reference point. This information not only caters to consumer preferences but also aids breeders in enhancing and creating new low GI cultivars. 

In this context, research efforts aimed at establishing correlations between rice quality parameters and starch digestibility characteristics are crucial for developing innovative predictive models. These models facilitate the swift classification of rice varieties based on their glycaemic index.

## Figures and Tables

**Figure 1 foods-13-00135-f001:**
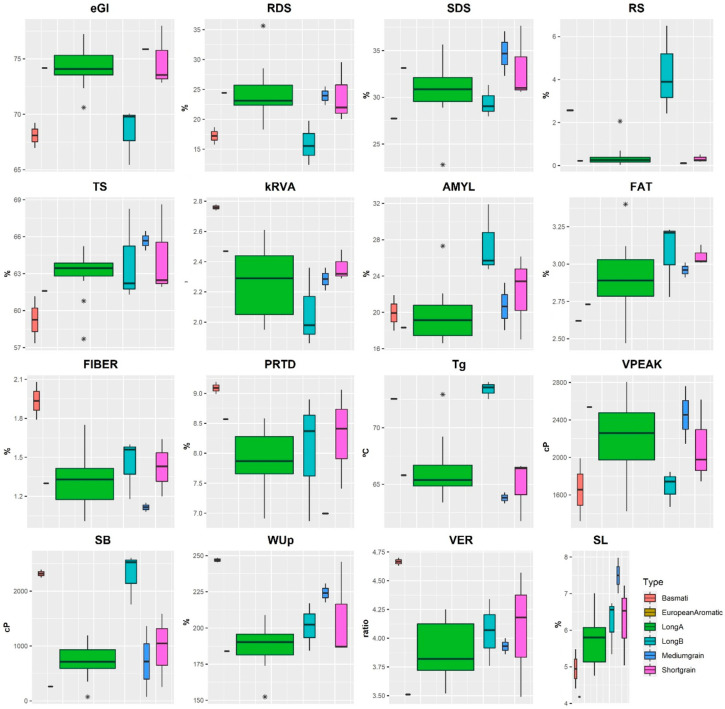
Boxplots of rice commercial types according starch digestion parameters, nutritional composition, cooking and viscosity properties. * RDS—rapidly digestible starch; SDS—Slowly digestible starch; RS—Resistant starch; TS—Total starch; k_RVA_—kinetic constant of starch hydrolysis RVA model; AMYL—amylose content; FAT—fat content; FIBER—fibre content; PRTD—protein content; Tg—Gelatinization temperature; VPEAK—Viscosity Peak; SB—Setback; WUp—Water uptake; VER—volumetric expansion ratio; SL—Solids leached.

**Figure 2 foods-13-00135-f002:**
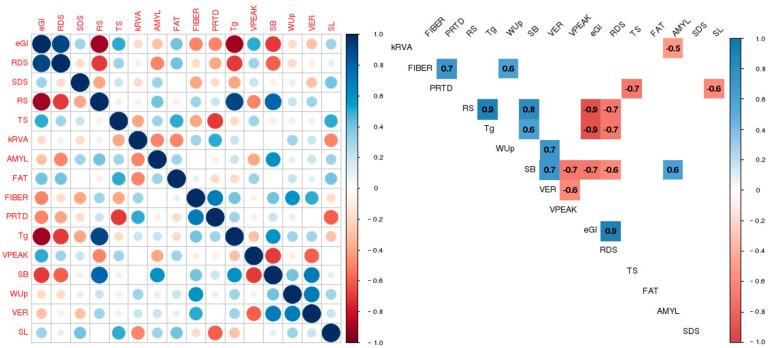
Significant (5% significance level) pairwise correlation coefficients for all the analysed parameters, with red indicating a negative correlation and blue denoting a positive correlation.

**Figure 3 foods-13-00135-f003:**
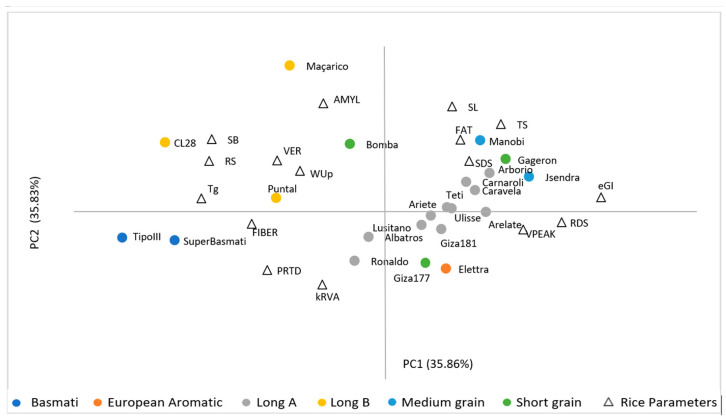
Principal component analysis.

**Figure 4 foods-13-00135-f004:**
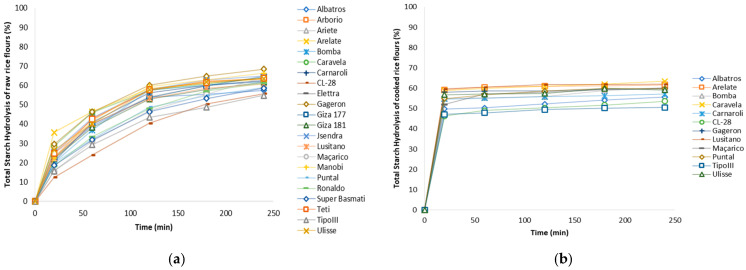
Total starch hydrolysis rate (%) in (**a**) different raw brown rice varieties; (**b**) cooked brown rice varieties.

**Figure 5 foods-13-00135-f005:**
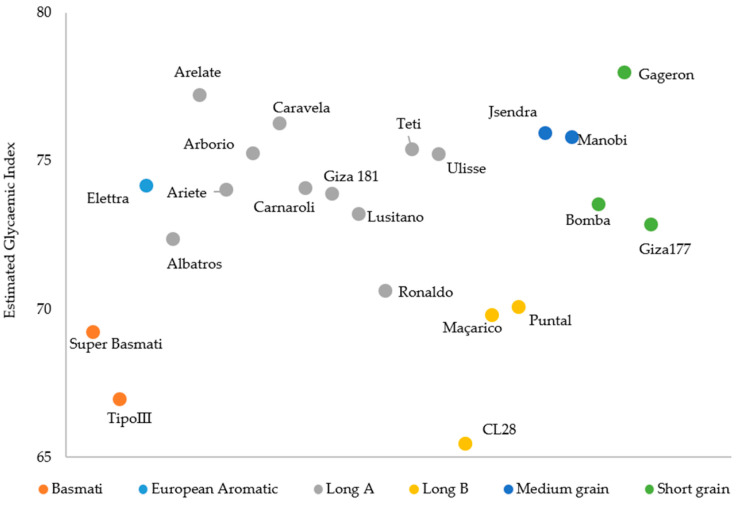
Estimated glycaemic index for different brown rice varieties.

**Table 1 foods-13-00135-t001:** Classification of rice varieties according to the commercial type based in the grain biometric parameters *.

Rice Variety	Type
Albatros	Long A
Arborio	Long A
Arelate	Long A
Ariete	Long A
Caravela	Long A
Carnaroli	Long A
Giza 181	Long A
Lusitano	Long A
Ronaldo	Long A
Ulisse	Long A
Teti	Long A
Bomba	Short grain
Gageron	Short grain
Giza 177	Short grain
CL28	Long B
Maçarico	Long B
Puntal	Long B
Jsendra	Medium grain
Manobi	Medium grain
Super Basmati	Basmati
TipoIII	Basmati
Elettra	European Aromatic

* Long A (length > 6.0 mm, length/width > 2 and <3), Long B (length > 6.0 mm; length/width ≥ 3), Medium grain (length > 5.2 mm and <6.0 mm, length/width < 3), Short grain (length ≤ 5.2 mm, length/width < 2).

**Table 2 foods-13-00135-t002:** Rice types means of estimated GI, Starch digestion parameters, nutritional parameters, viscosity and cooking parameters.

		Rice Types					
Parameters		Long A	Short Grain	Long B	Medium Grain	Basmati	European Aromatic
	eGI	74.33 ± 1.85 ^a^	74.79 ± 2.78 ^a^	68.44 ± 2.59 ^b^	75.88 ± 0.09 ^a^	68.1 ± 1.61 ^b^	74.17
Starch Digestion parameters	RDS (%)	24.52 ± 4.63 ^a^	23.86 ± 5.02 ^a^	15.89 ± 3.68 ^a^	23.95 ± 2.21 ^a^	17.24 ± 2.05 ^a^	24.43
	SDS (%)	30.76 ± 3.41 ^a^	33.08 ± 3.96 ^a^	29.44 ± 1.71 ^a^	34.69 ± 3.38 ^a^	27.73 ± 0.05 ^a^	33.14
	RS (%)	0.43 ± 0.57 ^c^	0.32 ± 0.18b ^c^	4.28 ± 2.06 ^a^	0.11 ± 0.01 ^bc^	2.57 ± 0.06 ^ab^	0.22
	TS (%)	62.88 ± 2.05 ^a^	64.35 ± 3.72 ^a^	63.93 ± 3.79 ^a^	65.67 ± 1.12 ^a^	59.25 ± 2.69 ^a^	61.60
	kRVA	2.26 ± 0.23 ^b^	2.36 ± 0.10 ^ab^	2.07 ± 0.26 ^b^	2.29 ± 0.11 ^ab^	2.76 ± 0.03 ^a^	2.47
Nutritional composition	AMYL * (%)	19.66 ± 3.10 ^b^	22.19 ± 4.69 ^ab^	27.46 ± 3.88 ^a^	20.65 ± 3.70 ^ab^	19.93 ± 2.77 ^ab^	18.31
	FAT (%)	2.89 ± 0.27 ^a^	3.05 ± 0.07 ^a^	3.07 ± 0.25 ^a^	2.96 ± 0.07 ^a^	2.62 ± 0.00 ^a^	2.73
	FIBER (%)	1.30 ± 0.21 ^b^	1.42 ± 0.22 ^ab^	1.45 ± 0.23 ^ab^	1.12 ± 0.05 ^b^	1.94 ± 0.21 ^a^	1.30
	PRTD (%)	7.89 ± 0.49 ^ab^	8.29 ± 0.83 ^ab^	8.05 ± 1.05 ^ab^	7.00 ± 0.01 ^b^	9.09 ± 0.14 ^a^	8.57
Viscosity parameters	Tg (°C)	66.24 ± 2.74 ^b^	64.93 ± 2.74 ^b^	73.37 ± 0.77 ^a^	63.81 ± 0.69 ^b^	72.53 ± 0.09 ^a^	65.80
	VPEAK (cP)	2202.09 ± 448.19 ^a^	2112.83 ± 451.22 ^a^	1687.33 ± 192.63 ^a^	2454.00 ± 434.16 ^a^	1655.75 ± 471.99 ^a^	2537.00
	SB (cP)	718.50 ± 321.29 ^b^	961.83 ± 671.22 ^b^	2295.83 ± 470.19 ^a^	718.50 ± 910.75 ^b^	2318.25 ± 100.76 ^a^	262.00
Cooking parameters	WUp * (%)	187.25 ± 15.32 ^b^	206.43 ± 34.14 ^ab^	201.24 ± 16.42 ^ab^	224.17 ± 9.19 ^ab^	246.94 ± 2.26 ^a^	283.93
	VER * (ratio)	3.88 ± 0.27 ^a^	4.08 ± 0.55 ^a^	4.06 ± 0.29 ^a^	3.93 ± 0.10 ^a^	4.66 ± 0.05 ^a^	3.51
	SL * (%)	5.72 ± 0.67 ^ab^	6.26 ± 1.11 ^ab^	6.21 ± 0.76 ^ab^	7.50 ± 0.69 ^a^	4.94 ± 0.76 ^b^	4.18

RDS—rapidly digestible starch; SDS—Slowly digestible starch; RS—Resistant starch; TS—Total starch; k_RVA_—kinetic constant of starch hydrolysis RVA model; AMYL—amylose content; FAT—fat content; FIBER—fibre content; PRTD—protein content; Tg—Gelatinization temperature; VPEAK—Viscosity Peak; SB—Setback; WUp—Water uptake; VER—volumetric expansion ratio; SL—Solids leached. Values in the same line with different letters (a, b and c) are significantly different at *p* ≤ 0.05 by HSD Tukey test (Note: The European aromatic type was not considered for the significant differences tests, because it has only one variety for observation (*n* = 1)); * parameters were determined in milled rice samples as outlined in the methods.

## Data Availability

Dataset obtained in this study were deposited in the Biodata.pt under the free accession link: https://doi.org/10.34636/DMPortal/5G5DA5.

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
