# Peer review of "Relationship between Physicochemical and Cooking Quality Parameters with Estimated Glycaemic Index of Rice Varieties"

_foods, 2023, doi:10.3390/foods13010135_

Round 1

Reviewer 1 Report

Comments and Suggestions for Authors

Reviewers' comments are noted in the manuscripts' pdf file

Author Response

We sincerely appreciate your valuable feedback and suggestions, which have proven to be immensely helpful in enhancing the quality of the article. Your insights on points 1 to 6 were carefully considered, and we would like to share the specific changes made in response to points 7 to 10:

  • 7-State the software used for ANOVA and PCA Calculations – Modified (Lines 208 and 217) with the appropriate citation were introduced.
  • 8-Results and Discussion sections should be merged into one integral section, since it is easier to track presented research- The Results and Discussion sections are integrated into a cohesive and easier-to-follow single section, as suggested.
  • 9-Table 2 needs formating. It is hard to track presented data – Formatted Table 2 for better clarity, addressing concerns about tracking presented data in the modified version (Lines 241-242).
  • 10- Reconsider rephrasing in the function of authors' results description and discussion or move to the introduction section. – Reformulated in the new presentation of results and discussion.

Cristiana Pereira

Reviewer 2 Report

Comments and Suggestions for Authors

This study evaluated the physicochemical and cooking quality parameters and estimated glycemic index of twenty-two rice varieties, containing a lot of work. The conclusion is that, resistant starch, gelatinization temperature and setback viscosity can be regarded as the predictors of rice starch digestibility and estimated glycemic index. The study is very interesting and useful to rice breeding and processing. Some suggestions:

1)     In the abstract, eGI should be noted. Long A and Long B rice varieties in the abstract and table 1 is not shown what are their differences. In table 1, the kernel length, width, and ratio of length and width, as well as moisture content, and contents of protein and amylose should be given, so that the readers can understand why they adopt these 22 rice varieties. Moreover, the data should be mean±SD, significant difference should be given.

2)     Table 2, the data should be given as mean±SD. Table 2 is not easy to read.

3)     In the abstract and conclusion sections, as for the effect of rice types, only the results of Long B and basmati rice were given, other rice varieties such as long A, short-grain, medium-grain, and European aromatic should be also given, so that this paper can be commonly read by the rice workers in east Asia, southeast Asia and so on.

4)     Why not to show the cooking time of 22 rice varieties?

Author Response

We deeply appreciate all the positive feedback and valuable suggestions, which proved to be highly beneficial in enhancing the article. The changes made in response to specific points, along with the corresponding line numbers, are detailed below:

1) In the abstract, eGI should be noted. Long A and Long B rice varieties in the abstract and table 1 is not shown what are their differences. In table 1, the kernel length, width, and ratio of length and width, as well as moisture content, and contents of protein and amylose should be given, so that the readers can understand why they adopt these 22 rice varieties. Moreover, the data should be mean±SD, significant difference should be given.

The material and methods section now includes the ranges for the biometrics of each type of rice, along with the key distinguishing factors (Lines 105- 113). Additionally, details on amylose and protein content information sourced directly from the raw data are accessible in the database https://doi.org/10.34636/DMPortal/5G5DA5 (Line 223-225).

2) Table 2, the data should be given as mean±SD. Table 2 is not easy to read.-

Table 2 has been updated to incorporate standard deviation and has been reformulated for improved readability. (Lines 242-243)

3)In the abstract and conclusion sections, as for the effect of rice types, only the results of Long B and basmati rice were given, other rice varieties such as long A, short-grain, medium-grain, and European aromatic should be also given, so that this paper can be commonly read by the rice workers in east Asia, southeast Asia and so on.

The following sentence was introduced in the conclusions section:

Long A, Short grain, and Medium grain types exhibit higher eGI, elevated viscosity, lower gelatinization temperatures, and resistant starch contents below 1%. Among them, Medium grain rice types displayed the least integrity during the cooking process. Reformulated (Lines 475-478)

4) Why not to show the cooking time of 22 rice varieties?

Indeed, various rice types demonstrate distinct cooking times. However, in the current study, we have employed a standardized cooking procedure with a fixed duration (20 min) (line 121) to quantify the reported cooking parameters.

Cristiana Pereira